# Optimize Electron Beam Energy toward In Situ Imaging of Thick Frozen Bio-Samples with Nanometer Resolution Using MeV-STEM

**DOI:** 10.3390/nano14090803

**Published:** 2024-05-05

**Authors:** Xi Yang, Liguo Wang, Victor Smaluk, Timur Shaftan

**Affiliations:** 1National Synchrotron Light Source II, Brookhaven National Laboratory, Upton, NY 11973, USA; vsmalyuk@bnl.gov (V.S.); shaftan@bnl.gov (T.S.); 2Laboratory for BioMolecular Structure, Brookhaven National Laboratory, Upton, NY 11973, USA

**Keywords:** electron bio-sample interaction, MeV-STEM, Monte Carlo simulation, beam broadening, Optics of STEM column

## Abstract

To optimize electron energy for in situ imaging of large biological samples up to 10 μm in thickness with nanoscale resolutions, we implemented an analytical model based on elastic and inelastic characteristic angles. This model has been benchmarked by Monte Carlo simulations and can be used to predict the transverse beam size broadening as a function of electron energy while the probe beam traverses through the sample. As a result, the optimal choice of the electron beam energy can be realized. In addition, the impact of the dose-limited resolution was analysed. While the sample thickness is less than 10 μm, there exists an optimal electron beam energy below 10 MeV regarding a specific sample thickness. However, for samples thicker than 10 μm, the optimal beam energy is 10 MeV or higher depending on the sample thickness, and the ultimate resolution could become worse with the increase in the sample thickness. Moreover, a MeV-STEM column based on a two-stage lens system can be applied to reduce the beam size from one micron at aperture to one nanometre at the sample with the energy tuning range from 3 to 10 MeV. In conjunction with the state-of-the-art ultralow emittance electron source that we recently implemented, the maximum size of an electron beam when it traverses through an up to 10 μm thick bio-sample can be kept less than 10 nm. This is a critical step toward the in situ imaging of large, thick biological samples with nanometer resolution.

## 1. Introduction

Driven by life-science applications, high-energy mega-electron-volt Scanning Transmission Electron Microscope (MeV-STEM) [1] could potentially break the fundamental limitation set by low-energy Electron Tomography (cryo-ET): the uncertainty and slow speed of Cryo-Focused Ion Beam slicing of large biological samples. This technique can produce just a few 300 nm thick lamellae per hour, and it often takes up to a day to obtain an intact 3D biological cell image [2,3,4,5]. Elastic and inelastic scattering of high-energy (greater than a few MeV) electrons with the unique combination of small characteristic angle and high penetration could turn the amplitude-contrast MeV-STEM into an appropriate microscope for sample thickness up to 10 µm with many applications in chemistry, biology, and life-science [1]. However, to minimize the angular broadening due to sample thickness and electron beam divergence, the probe beam must be focused on the specimen with a nanometre size and a milliradian semi-convergence angle. This requires a beyond state-of-the-art ultralow emittance electron source. For an MeV-STEM instrument, a high-energy (3–10 MeV) and high-brightness (a few picometer geometrical emittance, less than 10^−4^ normalized energy spread, and 1 nA beam current) electron sources are essential. Our recent progress on the MeV-STEM design [1] shows a beyond state-of-the-art electron source can be realized via two different approaches: (1) DC (direct current) gun [6,7], aperture, SRF (superconducting radio frequency) cavities, and STEM column; (2) CW (continuous wave) SRF gun [8,9,10], aperture, SRF cavities, and STEM column. Moreover, to mitigate the plural effects degrading the spatial resolution for large, thick biological samples, the electron beam energy must be boosted to 10 MeV or higher, depending on the sample thickness. As a result, despite where the electron beam being focused along the sample thickness dimension (e.g., top, middle, and bottom), the transverse size of the electron beam when it traverses through the sample can be kept ≤ 10 nm; thus, the size of the projected probe electron column in the STEM imaging mode can be minimized.

To achieve a resolution better than 10 nm for in situ imaging of large biological samples with a thickness of up to 10 μm, we implemented an analytical model based on the elastic and inelastic characteristic angles and benchmarked it by Monte Carlo (MC) simulations [11]. This model can be used to predict the transverse beam size broadening as a function of electron energy. To keep the beam size below 10 nm along its path in the sample, the electron energy is required to be 10 MeV or higher. Also, the effects of electron energy, beam lateral broadening, and dose limitation on resolution are explored. Since the main purpose of a high-energy MeV-STEM is to image frozen cells without cutting them into thin layers, the most striking and thickest organelle in frozen cells is the nucleus, which contains a dense fibrous network of DNA, RNA, and proteins, typically about 5–10 μm in diameter in various multicellular organisms [12]. Determining the ultimate resolution, especially for thick biological samples, is a complex problem which is beyond the scope of this manuscript. Also, regarding a specific sample thickness, further increasing the electron beam energy could not improve the overall resolution; instead, it just increases the cost and technical complexity of the instrument. In most life science applications where the sample thickness is less than 10 μm, there exists an ideal electron beam energy below 10 MeV for a particular thickness. However, for samples thicker than 10 μm, the optimal beam energy is preferred to be 10 MeV or higher, and it varies with the sample thickness. Moreover, as the sample thickness increases, the ultimate resolution may deteriorate.

## 2. Results

### 2.1. Analytical Model based on Electron Cross Sections

The angular distribution of scattering from a target atom can be described by differential scattering cross-section. In the Wentzel approximation, the differential cross-section for elastic scattering in the first-order Born approximation becomes [13]:(1)dσeldΩ=2ZR21+EE0aH(1+(θθel)2)2, θel=λ2πR , R=aHZ−13
where *σ_el_*, Ω, *Z*, *E*, *E*_0_, *a_H_*, *θ* and *λ* are the total elastic scattering cross-section, solid angle, atomic number, electron energy, rest energy of the electron, Bohr radius (0.0529 nm), scattering angle, and electron wavelength, respectively. *θ_el_* is the characteristic angle below which 50% of the electrons are elastically scattered. Integrating Equation (1) yields the total cross-section [13]:(2)σel≈h2c2Z43πE02β2 , β2=1−E0E+E02

The angle-dependent differential cross-section of inelastic scattering can be represented by the Bethe model [13,14]:(3)$$\frac{d\sigma_{inel}}{d\Omega }\approx \frac{Z\lambda^{4}{\left(1+\frac{E}{E_{0}}\right)}^{2}}{{4\pi^{2}a}_{H}^{2}}\left[\frac{1-{\left(1+\frac{\theta_{0}^{2}+\theta^{2}}{\theta_{0}^{2}}\right)}^{-2}}{{\left(\theta^{2}+\theta_{E}^{2}\right)}^{2}}\right]\;,\;\theta_{E}=\frac{∆E}{E}\frac{E+E_{0}}{E+{2E}_{0}}$$
where *σ_inel_*, *θ_E_*, and ∆*E* are the inelastic scattering cross-section, the angle determining the decay of the inelastic scattering, and mean energy loss from a single inelastic scattering event (e.g., 39.3 eV for amorphous ice [15]), respectively. An inelastic scattering is concentrated within much smaller angles than elastic scattering. Similarly, we define *θ_inel_* as the characteristic scattering angle, which represents the angle below which 50% of the electrons are inelastically scattered. This value is determined by numerically integrating the differential cross-section using Equation (3) (further explained in the following paragraph). The total cross-section for inelastic scattering can be estimated by [16,17].
(4)σinel≈1.5×10−6Z12β2ln⁡(2θc) , θc=ΔEβ2(E+E0)

The analytical model is derived based on the characteristic angles: elastic *θ_el_* and inelastic *θ_inel_*. These angles depend on the electron energy, and they can be obtained by numerically integrating the differential cross-sections (Equations (1) and (3)) azimuthally [1,13,14,15], then normalized by the total cross-section (Equations (2) and (4)), and finally, summed in the altitude dimension from 0 to π with a fine step (${∆\theta }_{alti}=0.001\;mrad$). As a result, for both *θ_el_* and *θ_inel_*, the angle corresponding to a 50% probability of the electron being scattered can also be obtained as the characteristic angle. Moreover, in the elastic scattering case, *θ_el_* can be accurately calculated either by Equation (1) or by the numerical integration method. The characteristic angles of elastic and inelastic scattering are shown in Figure 1a as functions of the electron energy.

The ratio of the total inelastic scattering cross-section and the total elastic scattering cross-section can be approximately expressed by Equation (5) [13,18]
(5)$$R_{in2el}=\frac{\sigma_{inel}}{\sigma_{el}}\approx \frac{\gamma }{Z}$$

Here, γ is a parameter close to 20 and hardly dependent on the atomic number or electron energy. This relationship holds for thin samples where multiple scattering is negligible and essentially all the high-angle elastic scattering is collected [13]. For example, in biological materials predominantly composed of the light elements carbon, nitrogen, and oxygen (Z=6, 7, 8), while one electron undergoes elastic scattering, approximately three electrons undergo inelastic scattering [13]. This ratio can be approximated to ~3, and the same ratio is applied in MC simulations. Since the scattering cross-section is a measure of the probability that a specific scattering process takes place and the ratio of elastic *θ_el_* and inelastic *θ_inel_* characteristic angles remains nearly constant across the electron energies ranging from 1 to 30 MeV (as detailed in Figure 1c), which aligns with $R_{in2el}≅3$, we can convert the normalized elastic and inelastic total cross-sections into corresponding weights in the ultimate scattering angular distribution. Thus, the effective critical angle, a weighted sum of elastic *θ_el_* and inelastic *θ_inel_* characteristic angles, can be estimated by Equation (6).
(6)$$\theta_{eff}(E)=\frac{R_{in2el}}{R_{in2el}+1}\theta_{inel}(E)+\frac{1}{R_{in2el}+1}\theta_{el}(E)$$

To avoid confusion, we deliberately named this weighted sum as the effective critical angle. Based on the characteristic angle of elastic (black) and inelastic (red) scattering shown in Figure 1a as the primary and secondary *y*-axis, we can obtain the effective critical angle as a function of electron beam energy (Figure 1b). Also, they are listed in Table 1.

For the ultimate resolution as a function of electron beam energy, one must take the following three factors into account: (1) angular broadening (AB) due to the semi-convergence angle and sample thickness; (2) emittance, diffraction, and aberration contribution (EC) to the focused beam size at the waist [1]; and (3) scattering broadening (SB) due to the angular distribution induced by all scattering channels, including both single and multiple elastic and inelastic scatterings. The ultimate beam size is the quadrature sum of the contributions from AB, EC, and SB:(7)$$\sigma_{tot}=\sqrt{{\sigma_{AB}}^{2}+{\sigma_{EC}}^{2}+{\sigma_{SB}}^{2}}$$
When a point electron is focused at the depth $t_{f}$ regarding the top surface of a specimen with thickness *T*, the AB $\left(\sigma_{AB}\right)$ of the electron probe at the sample thickness t is
(8)$$\sigma_{AB}\left(t\right)=\left|t-t_{f}\right|\cdot \tan\left(\alpha \right)≅\left|t-t_{f}\right|\cdot \alpha ,\;\mathrm{with}\;0\leq t,t_{f}\leq T$$
where α is beam semi-convergence angle. The focused beam size at the waist is dominated by the contribution of the geometrical emittance and aberrations, ${(\sigma }_{EC}≅1\;nm)$ with the demonstrated 2 pm geometrical emittance and $3\times 10^{-5}$ energy spread (see details in Section 2.5) [1]. Here, geometrical emittance is defined as the product of transverse size and semi-convergence angle at the beam waist. The aberrations include the contributions from both spherical and chromatic sources, dominated by chromatic aberration (see Section 2.5 for details). The SB ${(\sigma }_{SB})$ due to the effective critical angle $\theta_{eff}$ can be estimated via Equation (9)
(9)$$\sigma_{SB}\left(t\right)=a\cdot e^{b\cdot \left(\frac{t}{MFP}\right)}+c$$
where MFP is defined as the mean free path of the amorphous ice and *a*, *b*, and *c* are fitting parameters, which depend on $\theta_{eff}$ and need to be benchmarked by MC simulations. In MC simulations, the intensity change of the probe beam in each of these five categories (un-scattered, detected single elastically scattered, detected multiple elastically scattered, detected inelastic scattered, and undetected electrons) after passing through a sample slice of thickness *dt* has been described in details in our previous study [1] as well as implemented in the MC simulation code (see Section 2.4 for details). Consequently, the intensity change of the probe beam as it traverses through the sample is automatically considered in the simulations. It has been well studied by Ianik Pante [19] that the elastic and inelastic scattering cross-sections decrease with the increase in the electron energy and gradually reach saturation above the electron energy of 1 MeV with the *MFP* ≅ 0.5–1 μm (Figure 1d for inelastic scattering). Since we mainly focus on the high electron energy (> 1 MeV) case, we will assume a constant MFP(E)≅1 μm in the model. Any deviation of MFP can be taken care of by the fitting parameter *b*. We also assume when the electron beam traverses through the sample, a Gaussian intensity profile could be maintained; thus, the product of the peak intensity and the beam width stays constant with different sample thicknesses. The SB ${(\sigma }_{SB})$ can be approximated as exponentially increasing with the sample thickness (Equation (9)) up to 10 µm sample thickness since some early studies indicate that the peak intensity of an electron beam decreases exponentially with the sample thickness [1,13].

In summary, our goal is to predict how the transverse beam size changes with the depth of the probe beam as it traverses through the sample, with the electron beam energy higher than a few hundred kilo-electron-volts and the sample thickness exceeding a few micrometers using Equation (7). The AB $\left(\sigma_{AB}\right)$ and EC ($\sigma_{EC}$) terms can be completely determined by Equation (8), as well as the electron source and STEM column optics parameters (details in Section 2.5). Only fitting parameters *a*, *b*, and *c* in Equation (9) will be varied to ensure that the beam size, as a function of the sample depth predicted by Equation (9), agrees with the beam size simulated by the MC code (details in Section 2.4). The constraint applied to the fitting is that parameter *b* is linearly proportional to the effective critical angle. A standard least square method is used to find those three fitting parameters (*a*, *b*, and *c*). However, fully understanding the multiple elastic and inelastic scatterings, lateral intensity distribution, and achievable spatial resolution for large, thick biological samples up to 10 μm in thickness is a highly scientific challenge that requires significant resources and commitment. It is worth noting that these analytical formulas may be subject to limitations due to the above assumptions (e.g., sample thickness up to 10 μm and high electron energy exceeding several MeV), and an in-depth study of these complexities is beyond the scope of this paper.

The transverse beam size changes with the depth of the probe beam going into the sample. The maximum beam size could limit the ultimate resolution; thus, we treat the beam size as the resolution in the graphs shown in Figure 2. For the beam energy of 3 MeV (blue), 10 MeV (black), and 30 MeV (red), the relations between the beam size and the sample thickness are shown in Figure 2a,b with the probe beam focused on the middle and top of the sample, respectively. For the 3 MeV case, the resolution is dominated by the SB; once the sample thickness exceeds 2 μm (Figure 2c), the resolution, which is estimated via the maximum beam size in the sample thickness direction, does not depend on where the beam is focused anymore. Like Figure 2c, a different electron beam energy of 10 MeV is shown in Figure 2d.

In addition to where to focus the electron beam on the sample thickness dimension (see Figure 2a,b), it is required to have an electron beam energy of at least 10 MeV to achieve nanoscale resolution (≤10 nm) for the in situ imaging of large biological samples with the thickness up to 10 μm. However, further increasing the electron beam energy does not improve the resolution anymore.

### 2.2. Optimize Where to Focus on Different Sample Thicknesses

At each fixed electron beam energy (e.g., 3 MeV, 10 MeV, and 30 MeV), we explore three different cases where an electron probe is focused to the top, middle, and bottom along the sample thickness dimension. In the 3 MeV case (Figure 3a), the beam broadening is dominated by the SB, which is significantly worse than the combination of the AB and EC (named AB plus EC); thus, the maximum beam sizes are similar among those three cases despite where the beam is focused. In the 10 MeV case, the beam broadening is still dominated by the SB; however, the SB becomes closer to the AB plus EC, as shown in Figure 3b; among those three cases, the advantage of focusing the electron beam to the middle starts to show up. Further increasing the electron energy to 30 MeV (Figure 3c), the advantage of focusing the beam to the middle becomes pronounced. This is because in the high-energy (≥10 MeV) case, the AB plus EC dominates the beam broadening and can be mitigated by focusing on the middle of the sample. 

In summary, regarding a thick bio-sample, it does not matter where to focus the electron beam on the sample thickness dimension when the electron energy is below a few MeV (Figure 3a); however, there is a clear advantage when one focuses the electron beam to the middle of the sample once the electron energy exceeds 10 MeV (Figure 3b,c). The AB plus EC term (green curve in Figure 3d) is independent of the electron energy and mainly depends on the beam emittance and aberrations of the STEM column (see Section 2.5 for details). Instead, the SB contribution strongly depends on the electron energy, and it changes from slightly larger with the energy of 10 MeV (black solid in Figure 3d) to smaller with the energy of 30 MeV (red solid in Figure 3d) compared to the AB plus EC term (green curve in Figure 3d). 

### 2.3. Optimize Beam Energy for Different Sample Thickness and Image Contrast

We explore the relationship between the electron beam energy and the optimal sample thickness. So far, we assumed that the tolerance of the maximum projected beam size in the sample thickness dimension is 10 nm.
(10)$$\underset{0\leq t\leq T}{\max}\sigma_{tot}(t)\leq 10\;nm$$
For the sample thickness ≤ 10 μm, the electron beam energy can be optimized according to Equation (10) regarding each specific sample thickness, as shown in Figure 4a. However, if the sample is thicker than 10 μm and the beam energy exceeds 10 MeV, the AB plus EC could ultimately limit the resolution worse than 10 nm; thus, higher electron energy (≥10 MeV) could be necessary.

For example, regarding the optimal case where an electron probe is focused to half the thickness of the sample, the maximum size of the electron beam when it traverses through the sample vs. sample thickness is shown in Figure 4b, with six different electron energies, 3 MeV (blue), 10 MeV (black), 15 MeV (cyan dash), 20 MeV (cyan), 30 MeV (red), and 100 MeV (green dash), respectively. When the sample is thinner than 10 µm, the projected beam size can be kept below 10 nm with the electron energy of 10 MeV (black curve in Figure 4b); instead, when the sample thickness is thicker than 10 µm, up to 20 µm, the least electron energy of 30 MeV is required to keep the projected beam size below 10 nm (red curve in Figure 4b). However, when the beam energy exceeds 30 MeV, one could not reduce the maximum size (green dashed curve in Figure 4b) any further via purely increasing the electron beam energy since compared to the SB the AB plus EC becomes dominant, unless the beam emittance can be further mitigated. As shown in Figure 3d, AB plus EC (green solid line) is quite close to the total beam size of 30 MeV (red dashed line).

A dose-limited resolution (DLR) caused by radiation damage of imaging electrons can be estimated by Equation (11) [18,20]:(11)$$\delta_{Dose}=\sqrt{2}\cdot SNR\cdot \frac{1}{C\cdot \sqrt{DQE\cdot F\cdot Dc}}$$

Here, SNR, C, DQE, F, and Dc are the signal-to-noise ratio, contrast, detector quantum efficiency, fraction of electrons reached the detector (see details in Section 2.6), and characteristic electron fluence, respectively. In the case of STEM, SNR (signal-to-noise ratio) is much smaller compared to TEM (Transmission Electron Microscope). However, in line with the Rose criterion, which necessitates SNR≥3 for resolving features in an image, we assume SNR=3. Biological samples are particularly vulnerable to radiation damage. Therefore, the standard electron dose for cryo-ET typically ranges from 100 to 120 electrons per square angstrom (e^−^/Å^2^) divided among 40–60 images collected at various tilt angles. This averages to about 2 e^−^/Å^2^ per image. As a result, a characteristic electron fluence (Dc) of 100 e^−^/Å^2^ can be obtained by averaging the electron dose of 2 e^−^/Å^2^ per image across a tilting series of cryo-ET (e.g., 50 images with different tilt angles for frozen-hydrated samples) [13,21]. Also, a DQE (detector quantum efficiency) of 50% is routinely achievable for a direct electron detector [20].

As a high-energy MeV-STEM aims to eliminate the process of cutting an intact frozen cell into thin slices, the contrast of the frozen cell needs to be well examined for the estimation of the DLR. The density of diverse organelles and cells can have large variations, especially for large, thick, frozen biological specimens. For example, the average densities of protein, mitochondria, and bacteria E. coli are 1.3–1.4 g/ml [22,23], 1.19 g/ml [24], and 1.1 g/ml [25,26], and the density of the nucleus is about 1.4 g/ml [12]. Thus, these density variations are within the bound of 1.1 to 1.4 g/ml. Considering the presence of ice layers above the top and below the bottom surfaces of a frozen cell in conventional cryo-ET studies, the contrast of the biological sample embedded in amorphous ice can be determined to be 0.4 (set the DLR lower bound, called DLR LB) and 0.1 (set the DLR upper bound, called DLR UB) for nucleus and cell. The DLR for a 10 μm thick frozen biological sample is 1.5 nm for the nucleus. For other parts of a cell with a contrast of 0.1, the DLR for a 10-μm thick frozen biological sample is 6.0 nm. 

The DLR can be estimated for the upper bound with contrast C=0.1 (blue solid in Figure 4c) and the lower bound with C=0.4 (blue dash), whereas the fraction of electrons received by the detector can be obtained from Figure 7b (see later Section 2.6). Over the targeted sample thickness (1−10 μm, up to 20 μm), the DLR LB (blue dash) is smaller than the maximum projected beam size; thus, it likely will not limit the ultimate resolution. However, when the contrast is closer to 0.1, the DLR UB (blue solid) could become dominant and limit the resolution not better than 6 nm (Figure 4c). 

### 2.4. MC Simulation with Different Beam Energies

We benchmarked the beam size as a function of sample thickness predicted by the analytical model via a recently implemented MC simulation code [11,27,28,29,30,31,32] since in the study presented in this manuscript, accurately modelling inelastic scattering events in addition to elastic scattering events is extremely important, especially in such a low electron beam energy regime, ranging from a few hundred kilo-electron-volts to tens mega-electron-volts. Precisely calculating the differential scattering cross-section of electrons in amorphous ice to align with experimental observations is not so straightforward (see Section 2.1); thus, we implemented a new Monte Carlo code [11] for the purpose of precisely controlling parameters used to model inelastic scattering events in amorphous ice. 

The beam profile was generated based on the recently demonstrated state-of-the-art 2 pm geometrical emittance, $3\times 10^{-5}$ energy spread, and optimized chromatic (1.8 cm) and spherical (16 cm) aberrations of the STEM column (details in Section 2.5) [1]. An initial distribution with the RMS (root-mean-square) beam size of a nanometer and semi-convergence angle of a milliradian at the focal position and a total number of ten thousand electrons has been applied as the input to MC simulations. Then, an electron was randomly chosen from the electron profile. An event that includes elastically scattering, inelastic scattering, or no-scattering was selected based on the probability of every event in the thickness dt (chosen to be 0.5 nm) as defined by Equations (2) and (4). The total inelastic scattering cross-section ($\sigma_{inel}$) is scaled to be three times that of the total elastic scattering cross-section ($\sigma_{el}),$ $R_{in2el}\approx 3$ [13], as pointed out by Dr. Henderson [33], for 500 KeV electrons also without considering the effect of high voltages on the ratio. Moreover, this ratio has been applied to estimate the effective critical angle by Equation (6), which takes both the elastic and inelastic scatterings into account. When an electron interacts with the specimen, the chance of the electron undergoing event *i* (e.g., elastic scattering, inelastic scattering) within a sample thickness *dt* is given by
(12)$$P=\sigma_{i}\rho dt=K_{i}dt$$
where $\sigma_{i}$ is the cross-section for event *i*, and ρ is the sample density, $K_{i}$ is the scattering coefficient. The density of amorphous ice (0.92 g/cm^3^), as used by Langmore and Smith [16] and Jacobsen et al. [17], is applied in the simulation. The scattering angle was chosen based on the differential scattering cross-section for each angle interval *d*𝜃 for elastic (Equation (1)) and inelastic (Equation (3)) scattering events. Regarding elastic scattering, the cross-section for each angle interval is the sum of the oxygen cross-section ($\sigma_{O\_atom}$) and two times the hydrogen cross-section ($\sigma_{H\_atom}$), as $\sigma_{ice}=\sigma_{O\_atom}+2\sigma_{H\_atom}$. Regarding inelastic scattering, the cross-section is 110% of the oxygen cross-section [11]. The angle interval *d*𝜃 was set to 0.001 mrad for 𝜃 smaller than 0.6 mrad, 0.01 mrad for 𝜃 smaller than 6.3 mrad, 0.1 mrad for 𝜃 smaller than 67 mrad, 1 mrad for 𝜃 smaller than 600 mrad, and 10 mrad for other 𝜃 angles. The position of the electron was updated based on the scattering angle and the distance *dt* along the scattering direction. Simultaneously, the event type was recorded. The procedure was repeated for 10 μm/*dt* = 20,000 times. Since the scattering is circularly symmetric perpendicular to the incoming electrons, only a cross-section regarding the direction parallel to the incoming electrons was studied in the MC simulation. Each electron at depth *t* was classified into five groups (no scattering, single elastic scattering, multiple elastic scattering, at least one inelastic scattering, scattered outside the detector collection angles) [15,17] based on the events it experienced before reaching depth *t*. The beam size at depth *t* was calculated at a disc containing 68% of the electrons reaching depth *t*.

The comparisons are performed with the electron beam energies of 300 keV, 3 MeV, and 10 MeV with the standard least square method. There is a reasonably good agreement between the simulated (circles) [11] and analytically estimated (dashed line) beam sizes with the beam energy of 300 keV and sample thickness up to 1 μm (red), beam energy of 3 MeV and sample thickness up to 4 μm (blue), as well as beam energy of 10 MeV and sample thickness up to 5 μm (black), as shown in Figure 5a. Since a cross-section fixed to the direction parallel to the incoming electrons is considered in the MC simulation [11], the SB could be underestimated with the increase in the sample thickness. Hence, to extract the fitting parameters (*a*, *b*, and *c* in Equation (9)), we only consider MC simulated beam sizes [11] with the sample thickness up to a few MFP, e.g., 1 μm for the 300 keV case, a few microns for the beam energies of 3 MeV and 10 MeV. Moreover, we compare the beam size estimated via analytical formula (Equations (7)–(9)) and simulated via MC code up to 10 μm thickness with the optimal electron energy of 10 MeV. There is a reasonably good agreement between the analytical prediction (solid line) and MC simulation (circles) with the sample thickness in the range of 0 to 8 μm, as shown in Figure 5b, and the difference is still quite small, up to a 10 μm thickness. So, based on Equation (9), the fitting parameters $a≅1\;(nm),b≅10^{3}{\cdot \theta }_{eff}\left(E\right)(rad),\;and\;c≅0\;(nm)$ can be used to predict the beam size $\sigma_{SB}$ while it travels through the sample with a thickness of up to 10 μm and an electron energy exceeding a few mega-electron-volts.

### 2.5. Design STEM Column

It is well known that the column of a STEM is the reverse of that of a TEM from the sample to the detector. From our early study [1], the RMS size of the electron beam and normalized emittance ($\epsilon_{n}$) at the end of the electron source, which is shown as Aperture 2 in Figure 6, are roughly 0.5 μm and 10 pm, respectively. The emittance mentioned earlier is the geometric emittance $\left(\epsilon_{geo}\right)$, which is *γ*_0_(=$\frac{E}{m_{e}c^{2}}$ ~ 5–8, the Lorentz factor) times smaller than the normalized emittance $\epsilon_{n}$. It has been confirmed by our early study that the kinetic energies at the exit of the electron source can be varied in the range of 3.0–10.0 MeV [1]. Based on this electron source, a column consisting of two lenses, which are identical to the quintuplet designed by W. Wan [1,34], is chosen, with demagnification of 20 and 25, as shown in Figure 6 for the layout of the STEM instrument. 

From the result of our early study ^1^, the energy spread for the electron source is 100 eV. The normalized energy spread is $\frac{∆E}{E}=3.3\cdot 10^{-5}$. The final probe beam size at the waist is
(13)$$\sigma_{waist}=\sqrt{{\sigma_{d}}^{2}+{\sigma_{s}}^{2}+{\sigma_{c}}^{2}+{\sigma_{emit}}^{2}}$$
where σd=0.61λα, $\sigma_{s}=C_{s}\alpha^{3}/4$, $\sigma_{c}=\frac{C_{c}\alpha ∆E}{E}$, and $\sigma_{emit}=\frac{\epsilon_{geo}}{\alpha }$. Note that λ (=0.36 pm), α (=1 mrad), $C_{S}\;(=16\;cm)$, $C_{C}\;(=1.8\;cm)$, $\frac{\Delta E}{E}\;(=3.3\cdot 10^{-5})$, and $\epsilon_{geo}\;(=2\;pm\cdot rad)$ are the wavelength, aperture angle on the sample, spherical aberration, chromatic aberration, normalized energy spread, and geometrical emittance ^1^. The theoretically minimum beam waist size can be obtained by
(14)$$\delta_{th}=\sqrt{{\sigma_{d}}^{2}+{\sigma_{s}}^{2}+{\sigma_{c}}^{2}}$$
The theoretically minimum beam waist size $\left(\delta_{th,min}\right)$ is 0.52 nm when α=0.6 mrad ($\sigma_{d}=0.37\;nm$, $\sigma_{s}=0.01\;nm$, and $\sigma_{c}=0.36\;nm)$ [1]. The theoretical minimum beam size can always be reached through increasing demagnification if the current is of no concern. Thus, the beam waist size can be reasonably fixed to ~ 1.0 nm in the manuscript. 

### 2.6. Detector Signal at Different Electron Energy

The detector signal is defined as the fraction of incident electrons traversing through the entire sample (e.g., the amorphous ice) and being collected by the detector within a certain angle range (e.g., 0 to 1 mrad); see the schematic layout in Figure 7a. The detector signal calculated using the analytical model we implemented is shown in Figure 7b as a function of the electron beam energy for two different detector collecting angle ranges $\theta_{det}$: 0 to 1 mrad (red) and 0 to 10 mrad (black). The detector signal is estimated by integrating the electron distribution at the exit of the sample over a disc with the radius $R_{det}=T\cdot tan(\theta_{det})≅T\cdot \theta_{det}$. The normalized peak intensity on-axis at the detector is shown in Figure 7c as a function of sample thickness for three electron beam energies: 1 MeV (magenta), 3 MeV (blue), and 10 MeV (black). The detector signal includes all five channels: none interacted, single and multiple elastic and inelastic scattering electrons [1], within the angle of the detector ranging from 0 to 10 mrad.

## 3. Conclusions

We derived the analytical model based on the characteristic angles of elastic and inelastic scattering. The model has been benchmarked by MC simulations [11] and applied to explore the relationship between imaging resolution, which considers the maximum beam size inside the sample, sample thickness, and beam energy. As a result, while the sample thickness ≤10 μm, there exists an optimal electron beam energy below 10 MeV regarding each specific sample thickness. However, when the sample is thicker than 10 μm, the optimal beam energy should be 10 MeV or higher depending on the sample thickness, and the ultimate resolution could be worse with the increase in the sample thickness. Moreover, the dose-limited resolution has been examined in addition to the constraint of electron energy and beam broadening. Based on the scattering probability being proportional to the mass density, the above results can be adapted to different materials with a specific multiplier, which is the ratio of mass density between the targeting material and the amorphous ice used in the model.

## Figures and Tables

**Figure 1 nanomaterials-14-00803-f001:**
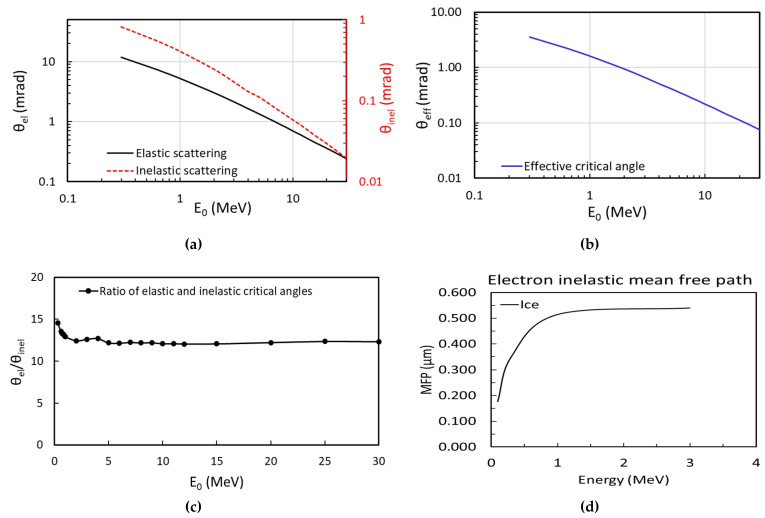
(**a**) Characteristic angles of elastic (black) and inelastic (red) as a function of electron beam energy are plotted as the primary and secondary *y*-axis. (**b**) The effective critical angle estimated by Equation (6) as a function of the electron beam energy is plotted. (**c**) Ratio of characteristic angles of elastic and inelastic scattering vs. beam energy. This ratio stays nearly constant in the energy ranging from 1 to 30 MeV. (**d**) *MFP* of inelastic scattering of amorphous ice as a function of electron energy.

**Figure 2 nanomaterials-14-00803-f002:**
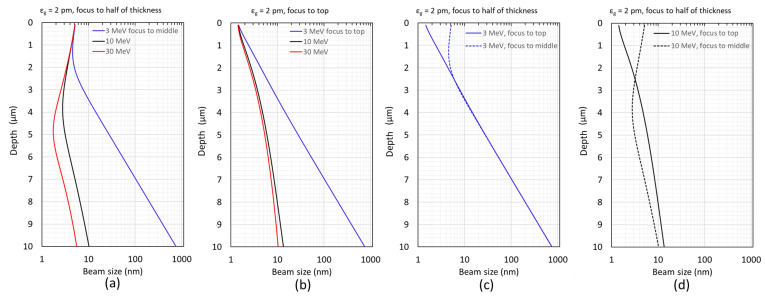
We estimate the beam size at three different beam energies (3 MeV in blue, 10 MeV in black, and 30 MeV in red) as the probe beam traverses through the sample thickness dimension. This estimation is based on Equation (7), taking into account the AB ($\sigma_{AB})$ and EC ($\sigma_{EC})$ terms, determined by the electron source and STEM column optics parameters (explained in Section 2.5), as well as the SB ($\sigma_{SB})$ term, validated by MC simulations (see details in Section 2.4). The beam size is analyzed when the probe beam is focused on (**a**) the middle or (**b**) the top. (**c**) For the 3 MeV case only, the beam size is plotted as solid and dashed blue curves when focused on the top and the middle, respectively. (**d**) Like (**c**) with a different electron beam energy of 10 MeV. In the plot titles, $\varepsilon_{g}$ denotes geometrical emittance.

**Figure 3 nanomaterials-14-00803-f003:**
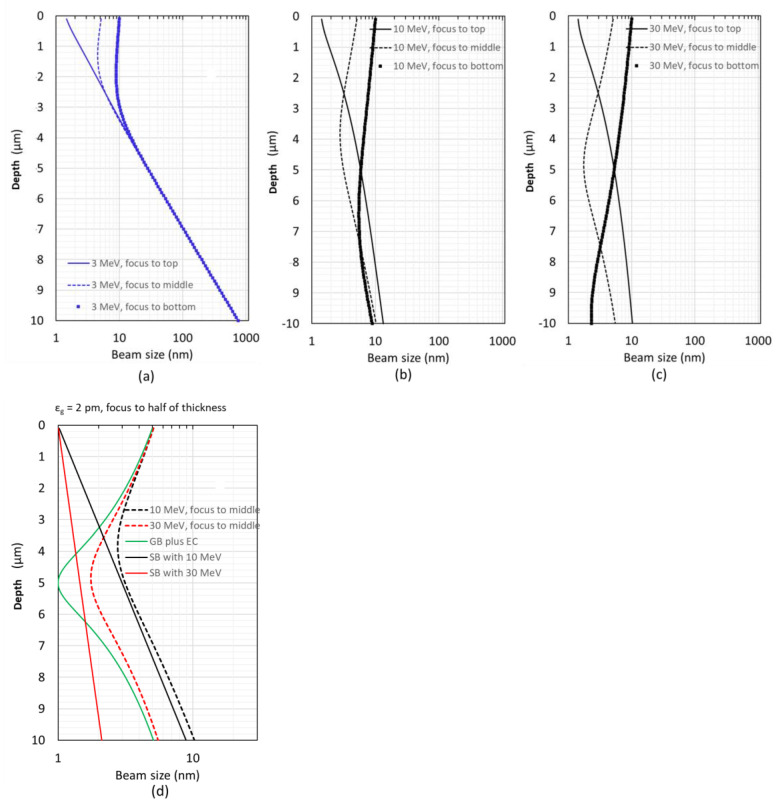
Regarding 3 different cases where an electron probe is focused to the top (solid), middle (dashed), and bottom (square) along the sample thickness dimension, we plot three different electron beam energies: (**a**) 3 MeV; (**b**) 10 MeV; (**c**) 30 MeV. (**d**) In case that an electron probe is focused to half the thickness of the sample, the AB plus EC term (green solid), SB terms with electron energy of 10 MeV (black solid) and 30 MeV (red solid), beam sizes with electron energy of 10 MeV (black dashed) and 30 MeV (red dashed) are plotted.

**Figure 4 nanomaterials-14-00803-f004:**
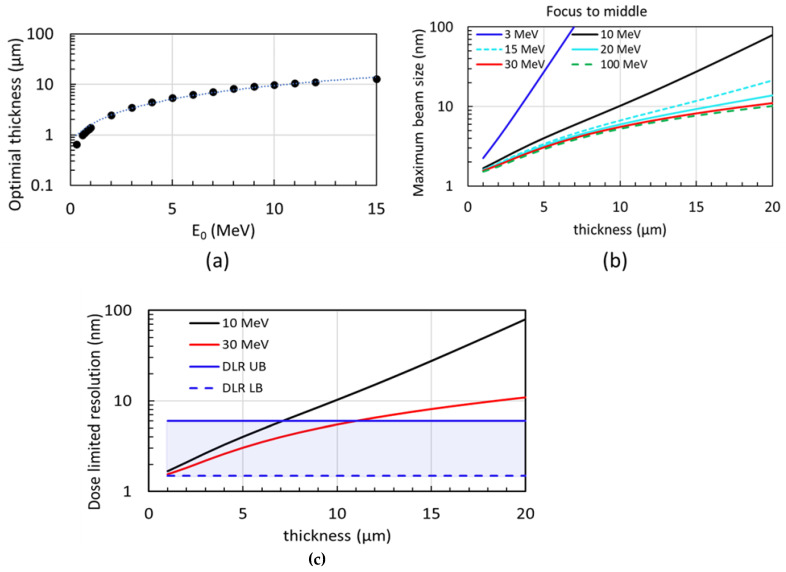
When the probe beam is focused on half of the sample thickness dimension, (**a**) electron beam energy determines the maximum (named optimal) sample thickness while the projected maximum beam size is kept below 10 nm with sample thickness up to 10 μm; (**b**) the maximum beam size when the electron beam traverses through the sample as a function of the sample thickness is shown in six different electron energies, 3 Mev (blue), 10 MeV (black), 15 MeV (cyan dash), 20 MeV (cyan), 30 MeV (red), and 100 MeV (green dash), respectively; (**c**) in addition to 10 MeV and 30 MeV cases in Figure 4 (**b**), the dose-limited resolution upper bound (DLR UB) with image contrast 0.1 and lower bound (DLR LB) with image contrast 0.4 are shown as the blue solid and dash curves, respectively (see later in detail). The fraction of electrons reached the detector within the collection angle of 0–10 mrad, and with the electron beam energy of 10 MeV or higher, can be approximated to one. Here, the image contrast (*C*) is defined as the density difference between the feature and the background, normalized by the density of the background.

**Figure 5 nanomaterials-14-00803-f005:**
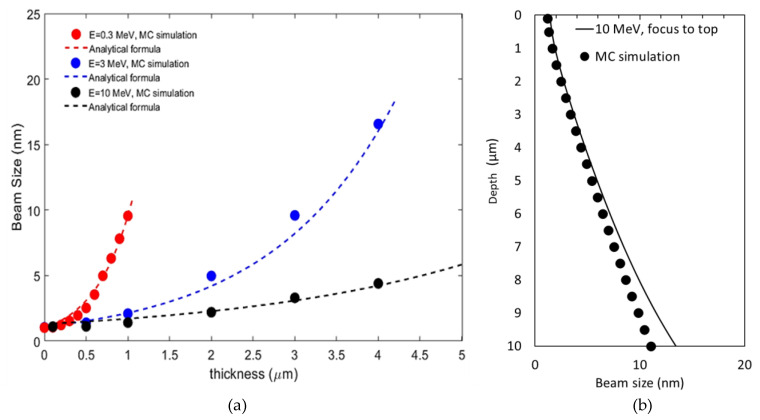
When the probe beam is focused on the top, (**a**) with the beam energy of 10 MeV, beam size vs. sample thickness up to 5 μm obtained via MC simulations is shown as black circles. The corresponding beam size estimated via analytical formula (Equation (7)) is plotted as the black dashed curve. Also, MC simulated (circles) and fitted (dashed lines) beam sizes vs. sample thickness up to 1.0 μm for beam energy of 0.3 MeV (red) and up to 4 μm for beam energy of 3 MeV (blue) are plotted. (**b**) Beam size vs. sample thickness predicted by the analytical formula Equation (7) (solid line) and simulated by MC code (circles) is plotted with the optimal electron beam energy of 10 MeV.

**Figure 6 nanomaterials-14-00803-f006:**
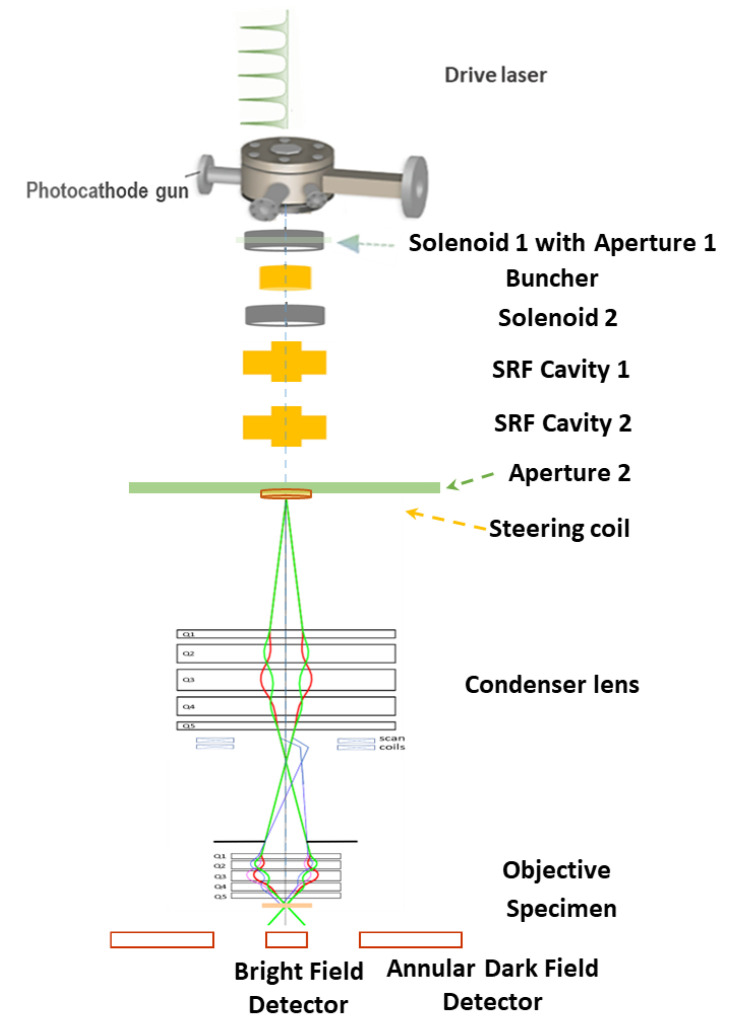
Schematic layout of the MeV-STEM instrument [1].

**Figure 7 nanomaterials-14-00803-f007:**
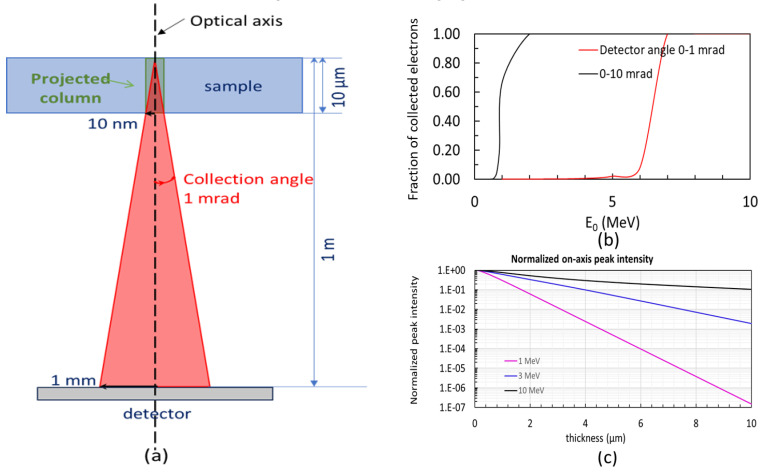
(**a**) Schematic of the probe beam traversing from the sample to the detector (e.g., detector angle relative to the optical axis and the top surface of the sample from 0 to 1 mrad). (**b**) Fraction of the collected electrons, which are normalized by the total number of incident electrons, as a function of the electron beam energy for two different detector angle ranges: 0 to 1 mrad (red) and 0 to 10 mrad (black). (**c**) Normalized peak intensity on-axis at the detector as a function of sample thickness is plotted at three different electron beam energies: 1 MeV (magenta), 3 MeV (blue), and 10 MeV (black).

**Table 1 nanomaterials-14-00803-t001:** Electron characteristic and effective critical scattering angles for H_2_O from 300 keV to 30 MeV. Here, $\theta_{el}$ and $\theta_{inel}$ are the characteristic angles for elastic and inelastic scattering, respectively. $\theta_{eff}$ is the effective critical angle.

Electron Energy (eV)	Elastic Characteristic Angle θ_el_ (mrad)	Inelastic Characteristic Angle θ_inel_ (mrad)	Effective Critical Angle θ_eff_ (mrad)
300,000 1000,000 3000,000	11.84	0.813	3.57
	5.24	0.406	1.61
	2.14	0.170	0.66
10,000,000 30,000,000	0.70	0.058	0.22
	0.24	0.020	0.07

## Data Availability

The datasets generated and analyzed during the current study are not publicly available due to the reason that we want to know who has an interest in our datasets, but they are available from the corresponding author upon reasonable request.

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
