# Peer review of "Optimize Electron Beam Energy toward In Situ Imaging of Thick Frozen Bio-Samples with Nanometer Resolution Using MeV-STEM"

_nanomaterials, 2024, doi:10.3390/nano14090803_

Round 1

Reviewer 1 Report

Comments and Suggestions for Authors

Apparently, the presented manuscript is a continuation of the work described in the paper X.Yang, et al., Towards Construction of a Novel Nanometer-resolution MeV-STEM for Imaging of Thick Frozen Biological Samples. Photonics 11(3), 252, doi:10.3390/photonics11030252 (2024)

The authors provide detailed calculations related to determining the optimal conditions for obtaining in-situ images when studying thick samples on a high-voltage STEM.

I would like to give just a few recommendations to the authors of this manuscript:

1) there is no need to insert any references in the Abstract section;

2) I believe that it is worth adding either in the title of the manuscript or at the very beginning of the Abstract for which type of microscopes the calculations presented in this work were made;

3) Line 414: please remove extra text from ref.1.

4) references 19 and 22 appear to be the same.

In my opinion, the manuscript can be published in the journal after minor revisions.

Author Response

Dear Reviewer,

we appreciated the reviewer spending valuable time in reviewing our paper and being positive on the continuation of our previous work - determining the optimal conditions for in-situ imaging large/thick bio-samples via the proposed MeV-STEM instrument.

We made all those changes, which greatly improved the quality of our manuscript.

Thank you so much.

Reviewer 2 Report

Comments and Suggestions for Authors

The authors discuss the need for a very small probe for STEM to allow the traversal of a thick biological specimen with little broadening of the probe. I find that many details are not provided and many terms are not well explained. In particular, the way the Monte Carlo simulations were done are not explained. In addition, there should be a clear distinction between depth in the specimen and its thickness. This is an interesting manuscript but should be improved considerably before publication.

Line 67:

"complicity" should be "complexity"

Line 67:

"Hence, the optimal selection of the electron beam energy for in-situ imaging of large thick biological samples with a thickness up 68 to 10 𝜇m is 10 MeV."

This sentence is very abrupt. The authors should briefly state if this is a conclusion that they reached or is broadly accepted.

Line 88:

I do not see where theta-inel is given in an equation. I presume it should be in equation 3. Or is it just an operation variable to be determined?

Line 108:

If gamma is close to 20 and Z=8, the ratio from equation 5 should be 2.5, not 3. Please explain.

Line 110:

What is the difference between "characteristic" and "critical" angles? Are these interchangeable terms? If so, it would be better to use only one term.

Line 115:

I don't understand why an effective critical angle is required. If the aim is to collect for both elastic and inelastic scattering, the range of collection angles is of importance.

Line 133:

It would be good to describe "geometrical emittance" here. I presume the "aberrations" referred to here is mostly spherical. The text would benefit from better stating which aberrations are influential.

Line 137:

How are the parameters a, b and c determined for the simulation? If it is through fitting, how is that done?

Lin 167:

"transverse beam size"

I presume this is at the specimen exit face. Please clarify.

Figure2:

"Thickness" typically refers to the total specimen thickness. The graphs would be less confusing if the y axis is labeled "Depth".

What is epsilon-g?

I'm also confused as to where the data for Figure 2 comes from. Are these the results of simulation? Or is it from equation 7? The figure legend is also not clear on the purpose of the graphs. What particular point should the reader understand?

Line 180:

"It is evident in Fig. 3 that the optimal choice of electron beam 180 energy for in-situ imaging of a large thick (up to 10 μ m) bio-sample is 10 MeV."

I'm not sure I understand how the authors reached this conclusion. The "optimal" solution is to use an accelerating voltage as high as possible, not just 10MeV.

Figure 3:

What is the difference between Figure 2a and 3? They seem to be the same parameters but different results.

Author Response

Dear Reviewer,

We appreciated the reviewer spending valuable time in reviewing our paper and giving critically important comments and suggestions on our manuscript. We made all those changes, which greatly improved the quality and readability of our manuscript.

Thank you so much.

Reviewer 3 Report

Comments and Suggestions for Authors

This article investigates the technical characteristics of a new instrument, MeV-TEM, for biological applications. The article is of great interest as the proposed instrument can be a valid alternative to the more classical cryo-TEM and similar instrumentation. The study focuses on the energy dependence of the beam size.

The methodology employed seems interesting, but I do not understand why it was necessary to develop a new Monte Carlo code when tools like Geant4 are available that can easily simulate the described phenomena. It would be useful to motivate the choice, even if the article only uses it for comparison with an analytical approach. The advantage of using codes like Geant4 is that they are "certified" for the quality of the results.

Another point that leaves me a bit perplexed is the calculation of the contrast in the paragraph dedicated to the dependence of the resolution on the dose. I do not understand on what basis the absorption is determined. In the sense that it depends on the composition of the material in an aqueous matrix and therefore on the relative material attenuation coefficient, which is also related to its size. How was this estimate made? Furthermore, I did not understand if DLR is d dose, which should be clarified better. In general, the influence of the composition should be discussed a bit instead of being simply described as complicated, especially above 10 micrometers of  thickness.

 A minor comment concerns the name of the ordinate in Figure 8b: although the meaning is intuitive by reading the caption, I think it should be modified to align with the caption. Moreover, I didn’t understand why sSB(t) is set to 0.0 nn in the caption of figure 3

 In conclusion, the article is interesting and well-written, and by addressing the comments on facts, it is certainly acceptable for publication.

Author Response

Dear Reviewer,

We appreciated the reviewer spending valuable time in reviewing our paper and giving critically important comments and suggestions on our manuscript. We made all the changes, which greatly improved the quality and readability of our manuscript. 

Thank you so much.
